# Twelve phosphomimetic mutations induce the assembly of recombinant full-length human tau into paired helical filaments

Sofia Lövestam, Jane L Wagstaff, Taxiarchis Katsinelos, Jenny Shi, Stefan MV Freund, Michel Goedert*, Sjors HW Scheres*

MRC Laboratory of Molecular Biology, Cambridge, United Kingdom

## eLife Assessment

This manuscript describes the identification and characterization of 12 specific phosphomimetic mutations in the recombinant full-length human tau protein that trigger tau to form fibrils. This **fundamental** study will allow in vitro mechanistic investigations. The presented evidence is **convincing**. This manuscript will be of interest to all scientists in the amyloid formation field.

*For correspondence:
mg@mrc-lmb.cam.ac.uk (MG);
scheres@mrc-lmb.cam.ac.uk
(SHWS)

## Abstract

The assembly of tau into amyloid filaments is associated with more than 20 neurodegenerative diseases, collectively termed tauopathies. Electron cryo-microscopy (cryo-EM) structures of brain-derived tau filaments revealed that specific structures define different diseases, triggering a quest for the development of experimental model systems that replicate the structures of disease. Here, we describe 12 phosphomimetic serine/threonine-to-aspartate mutations in tau, which we term PAD12, that collectively induce the in vitro assembly of full-length three-repeat tau into filaments with the same structure as paired helical filaments extracted from the brains of individuals with Alzheimer's disease. Solution-state nuclear magnetic resonance spectroscopy suggests that phosphomimetic mutations in the carboxy-terminal domain of tau may facilitate filament formation by disrupting an intramolecular interaction between two IVYK motifs. PAD12 tau can be used for both nucleation-dependent and multiple rounds of seeded assembly in vitro, as well as for the seeding of tau biosensor cells. PAD12 tau can be assembled into paired helical filaments under various shaking conditions, with the resulting filaments being stable for extended periods of time. They can be labelled with fluorophores and biotin. Tau filaments extracted from the brains of individuals with Alzheimer's disease have been known to be made of hyperphosphorylated and abnormally phosphorylated full-length tau, but it was not known if the presence of this post-translational modification is more than a mere correlation. Our findings suggest that hyperphosphorylation of tau may be sufficient for the formation of the Alzheimer tau fold. PAD12 tau will be a useful tool for the study of molecular mechanisms of neurodegeneration.

## Introduction

The assembly of tau into amyloid filaments characterises a group of neurodegenerative diseases that are called tauopathies. Alzheimer's disease (AD) is the most common tauopathy. Whereas in most tauopathies, tau is the only protein that assembles into filaments, in AD, intracellular assemblies of tau co-exist with extracellular plaques of filamentous amyloid-β.

Six tau isoforms, with lengths ranging from 352 to 441 amino acids, are expressed in the adult human brain by alternative mRNA splicing from a single gene (*MAPT*). The tau sequence (*Figure 1A*) comprises an amino-terminal projection domain (residues 1–150), a proline-rich region (residues

**Figure 1.** Electron cryo-microscopy (cryo-EM) analysis of PAD12 tau filaments. (**A**) Schematic of PAD12 tau. The amino-terminal inserts N1 (residues 44–73) and N2 (74–102) are shown in grey, the proline-rich region (151–243) is in light grey, the microtubule-binding repeats are in purple (R1, 244–274), blue (R2, 275–305), green (R3, 306–336), and yellow (R4, 337–368), and the carboxy-terminal domain (369–441) is shown in orange. The 12 phosphomimetic mutations of PAD12 are indicated with vertical lines. (**B**) Cryo-EM micrograph of paired helical filaments (PHFs) composed of a 1:1 mixture of 0N3R and 0N4R PAD12 tau. Arrows show PHFs (blue) and single protofilaments with the Alzheimer tau fold (light blue). (**C**) Cross-sections of cryo-EM reconstructions perpendicular to the helical axis, with a thickness of approximately 4.7 Å for assembly reactions with 0N3R PAD12 tau, 0N4R PAD12 tau, and a 1:1 mixture of 0N3R and 0N4R PAD12 tau. Inserts show pie charts with the particle distribution per filament types (PHFs in blue; single protofilaments with the Alzheimer fold in light blue; single protofilaments with the chronic traumatic encephalopathy [CTE] fold in yellow; discarded solved particles [coloured] and discarded filaments in grey). (**D**) Cryo-EM density map of the 1:1 0N3R:0N4R PAD12 tau mixture in transparent grey, superimposed with its refined atomic model. (**E**) Main chain trace of the atomic model shown in D (blue), overlaid with the model of PHFs from Alzheimer's disease (AD) brain (PDB-ID: 6hre) (grey).

The online version of this article includes the following source data and figure supplement(s) for figure 1:

**Figure supplement 1.** Purification of 0N3R and 0N4R PAD12 tau.

**Figure supplement 1—source data 1.** Original gels annotated from *Figure 1—figure supplement 1*.

**Figure supplement 1—source data 2.** Original gels from *Figure 1—figure supplement 1*.

**Figure supplement 2.** Fourier shell correlation curves.

**Figure supplement 3.** Electron cryo-microscopy (cryo-EM) analysis of tau297–391 and PAD12 tau paired helical filaments (PHFs).

**Figure supplement 4.** Quantification of PAD12 tau filament types with time.

**Figure supplement 5.** Reproducibility of paired helical filaments (PHF) formation.

151–243), four microtubule-binding repeats (R1-R4; residues 244–368), and a carboxy-terminal domain (residues 369–441). Tau isoforms vary by the incorporation of zero, one, or two inserts of 29 amino acids at the amino-terminal domain (0N, 1N, or 2N isoforms) and the presence or absence of the second microtubule-binding repeat, resulting in three- or four-repeat (3R or 4R) tau isoforms (*Goedert et al., 1989*). A mixture of all six tau isoforms is present in AD and chronic traumatic encephalopathy (CTE), whilst only 3R tau assembles in Pick's disease, and only 4R tau assembles in progressive supranuclear palsy (PSP), corticobasal degeneration (CBD), globular glial tauopathy (GGT), and argyrophilic grain disease (AGD).

Electron cryo-microscopy (cryo-EM) revealed that specific folds of assembled tau define different tauopathies (*Shi et al., 2021*). Residues from the microtubule-binding repeats plus approximately 10 residues from the carboxy-terminal domain are ordered in tau filaments from the different diseases. In tau filaments from AD and several other diseases, the remaining residues are disordered, forming a fuzzy coat that surrounds the ordered core (*Wischik et al., 1988*). Intriguingly, with the exception of a familial tauopathy caused by a P301T mutation in tau (*Schweighauser et al., 2025*), only a single tau fold has been observed for each disease, whereas multiple individuals with a given disease always had the same filaments. The consistent structure of tau filaments across individuals with the same

tauopathy supports the prion hypothesis (*Prusiner, 1998*), in which tau filaments propagate their structure through templated misfolding.

The specific tau folds that form and propagate in the human brain may be influenced by the cellular environments where they assemble. Model systems used to study tauopathies should therefore replicate the same tau folds observed in diseased human brains. However, this has proven challenging. For instance, when recombinant wild-type full-length tau is assembled in vitro, it requires the addition of negatively charged molecules (*Goedert et al., 1996*; *Kampers et al., 1996*; *Pérez et al., 1996*; *Wilson and Binder, 1997*), but the structures of tau filaments assembled in the presence of heparin or RNA are unlike those extracted from human brains (*Abskharon et al., 2022*; *Lövestam and Scheres, 2022*; *Zhang et al., 2019*). Similarly, attempts to reproduce disease-specific tau filaments by over-expressing full-length human tau in SH-SY5Y cells, combined with seeding using human brain-derived tau filaments, did not recapitulate the structures of AD or CBD (*Tarutani et al., 2023*). Additionally, mice over-expressing 4R tau with the P301L or P301S mutation under the control of different promoters form tau filaments with different structures, suggesting that different promoters may cause filament assembly in distinct cell types, affecting which tau fold forms (*Schweighauser et al., 2023*; *Zhao et al., 2024*). None of the tau filament structures from mouse models were identical to those observed in human brains.

We previously reported that, under shaking conditions, the truncated tau construct comprising residues 297–391 (tau297–391) (*Al-Hilaly et al., 2020*; *Novak et al., 1993*) assembles into paired helical filaments (PHFs), which are the main filament type in AD (*Lövestam et al., 2022*). The addition of sodium chloride to the assembly reaction of tau297–391 led to the formation of CTE filaments. Subsequent time-resolved cryo-EM studies revealed that formation of the disease-specific structures occurs through many intermediate amyloid filaments, with a common first intermediate amyloid (FIA) observed in the assembly of AD and CTE filaments (*Lövestam et al., 2024*). But unlike filaments isolated from AD brains which contain full-length tau (*Lee et al., 1991*), filaments of tau297–391 lack the fuzzy coat that is typical of tau filaments inside brain cells.

Besides the cellular environment in which they assemble, different tau folds may also be determined by chemical modifications of tau itself. PHF-tau from the brains of individuals with AD is hyperphosphorylated and abnormally phosphorylated (*Grundke-Iqbal et al., 1986*; *Iqbal et al., 2016*); this is believed to precede filament assembly to which it has been linked. Hyperphosphorylation of tau may also accelerate aggregation indirectly by detaching tau from microtubules. Mass spectrometry and epitope mapping of antibodies that were raised against PHFs from AD brains (*Greenberg et al., 1992*; *Mercken et al., 1992*) led to the identification of multiple sites, all in the fuzzy coat of tau, that are hyperphosphorylated in PHFs from AD brains (*Lee et al., 2001*). Monoclonal antibody AT270 recognises phosphorylation of threonine 181 (T181); AT8 recognises phosphorylation of serine 202 (S202) and threonine 205 (T205), AT100 recognises phosphorylation of T212, S214, and T217, AT180 recognises phosphorylation of T231 and S235, and PHF-1 recognises phosphorylation of S396 and S404. A comprehensive mass spectrometry map of post-translational modifications confirmed that these residues are hyperphosphorylated in assembled tau from AD brains (*Wesseling et al., 2020*).

Based on these observations, we previously designed four phosphomimetic mutations in the carboxy-terminal domain of tau (S396D, S400D, T403D, and S404D) that allowed the assembly of recombinant tau297–441 into filaments with the Alzheimer fold, albeit only with a single protofilament (*Lövestam et al., 2022*). A similar construct was also shown to be suitable for the production of large amounts of isotope-labelled protein for nuclear magnetic resonance (NMR), but again it yielded filaments with the Alzheimer protofilament fold, but with protofilament packings distinct from those in PHFs (*Duan et al., 2024b*). Additionally, a tau construct with phosphomimetic mutations in the proline-rich region (S202E, T205E, S208E), as well in the carboxy-terminal region (S396E, S400E, T403E, S404E), was found to assemble, but the filaments formed were again distinct from those found in AD (*Mammeri et al., 2024*).

We hypothesised that the introduction of additional phosphomimetic mutations in the proline-rich region of tau might facilitate the assembly of recombinant full-length tau. Based on the epitopes of antibodies and the mass spectrometry data described above, we mutated eight residues within the proline-rich region of tau (T181, S202, T205, T212, S214, T217, T231, S235) to aspartate. Combined with the four previously described mutations in the carboxy-terminal domain of tau, this resulted in twelve phosphomimetic mutations in full-length tau, nine of which were in serine/threonine-proline

**Table 1.** Filament assembly conditions.

| Construct | Concentration (µM) | Buffer | Shaking | Time (days) | Fold |
|---|---|---|---|---|---|
| 0N3R PAD12 | 100 | 100 mM KPhos*, 10 mM TCEP | 500 rpm, 1 min on:1 min off | 7 | AD |
| 0N4R PAD12 | 100 | 100 mM KPhos, 4 mM TCEP | 500 rpm, 1 min on:1 min off | 7 | CTE-singlet |
| 0N3R:0N4R | 50:50 | 100 mM KPhos, 10 mM TCEP | 500 rpm, 1 min on:1 min off | 7 | AD |
| tau0–391 | 100 | 100 mM KPhos, 10 mM TCEP | 500 rpm, 1 min on:1 min off | 7 | AD |
| tau151–391 | 100 | 100 mM KPhos, 10 mM TCEP | 500 rpm, 1 min on:1 min off | 7 | AD |
| 0N3R PAD12 (freeze-thaw) | 100 | 100 mM KPhos, 10 mM TCEP | 500 rpm, 1 min on:1 min off | 7 | AD |
| 0N3R PAD12 (tube) | 100 | 100 mM KPhos, 10 mM TCEP | 500 rpm | 7 | AD |
| 0N3R PAD12 (+seeds from 0.8 µg AD brain) | 40 | 20 mM HEPES at pH 7.3 and 4 mM KPhos at pH 7.2, plus 300 mM sodium citrate and 4 mM TCEP | 500 rpm, 1 min on:1 min off | 2 | AD |
| 0N3R PAD12 | 50 | 100 mM KPhos, 10 mM TCEP | 500 rpm, 1 min on:1 min off | 7 shaking; 14 quiescent | AD |
| 0N3R PAD12 | 50 | 20 mM HEPES at pH 7.3 and 4 mM KPhos at pH 7.2, plus 300 mM sodium citrate and 4 mM TCEP | 500 rpm, 5 s on:5 s off | 7 shaking; 14 quiescent | AD |
| tau297–441 Δ392–395 | 50 | 100 mM KPhos, 4 mM TCEP | 500 rpm, 1 min on:1 min off | 3.17 | New |

*KPhos is potassium phosphate.

motifs (*Figure 1A*). We show that these mutations, which we refer to as twelve phosphomimetics of AD, or PAD12, allow both nucleation-dependent and seeded assembly of full-length recombinant tau into PHFs, thereby opening new avenues for studying the role of tau's fuzzy coat in the molecular mechanisms of disease. Our findings suggest a mechanism by which phosphorylation in the fuzzy coat of tau leads to filament formation, and they indicate that the hyperphosphorylation of tau at specific sites may be sufficient for the formation of the Alzheimer tau fold from full-length protein.

## Results

### Assembly of PAD12 full-length tau into PHFs

0N3R and 0N4R versions of PAD12 tau were expressed in *Escherichia coli* and purified, followed by in vitro assembly under shaking conditions (*Figure 1A*; *Figure 1—figure supplement 1*; *Table 1*; Materials and methods). Assembly was carried out with 500 rpm orbital shaking (2 min on; 1 min off) for 7 days at a protein concentration of 100 µM in a buffer containing 100 mM potassium phosphate at pH 7.2, 400 mM potassium citrate, and 4 mM tris(2-carboxyethyl) phosphine (TCEP). We used cryo-EM to determine the structures of the resulting filaments (*Figure 1B–D*; *Figure 1—figure supplement 2*; *Table 2*). 0N3R-PAD12 tau primarily formed PHFs (85%), with a minor population (15%) of single protofilaments with the Alzheimer fold. The remainder of the filaments, including false positives from automated filament picking (*Lövestam and Scheres, 2022*), were discarded during image processing. 0N4R-PAD12 tau assembled into filaments consisting only of a single protofilament with the CTE fold. It is possible that small amounts of sodium chloride in the protein sample led to the formation of protofilaments with the CTE fold. The observation that part of the second microtubule-binding repeat is ordered, forming cross-beta packing against the third repeat, may be related to this construct forming filaments with only a single protofilament.

Because tau filaments from AD brains contain equimolar amounts of 3R and 4R tau (*Goedert et al., 1992*), we assembled a 1:1 mixture of 0N3R-PAD12 and 0N4R-PAD12 tau under identical conditions.

**Table 2.** Electron cryo-microscopy (cryo-EM) statistics.

| LMB Krios G4 | PAD12 0N3R:0 N4R (EMDB-51884) (PDB-9H5G) | PAD12 0N3R AD-seeded (EMDB-51886) (PDB-9H5J) | 297–441 Δ392–395 (EMDB 54485) (PDB-9S2B) |
|---|---|---|---|
| **Data acquisition** | | | |
| Electron gun | CFEG | CFEG | FEG |
| Detector | Falcon 4i | Falcon 4i | Falcon 4i |
| Energy filter slit (eV) | 10 | 10 | na |
| Magnification | 165,000 | 165,000 | 96,000 |
| Voltage (kV) | 300 | 300 | 300 |
| Electron dose (e-/Å$^2$) | 40 | 40 | 35 |
| Defocus range (μM) | 0.5–2.5 | 0.5–2.5 | 0.5–2.5 |
| Pixel size (Å) | 0.744 | 0.744 | 0.824 |
| **Data processing** | | | |
| Initial particle images (no.) | 664,772 (Autopick) | 163,335 (manual) | 643,712 (Autopick) |
| Final particle images (no.) | 56,351 | 125,334 | 213,971 |
| Helical twist (°) | 179.47 | 179.39 | −2.9 |
| Helical rise (Å) | 2.43 | 2.42 | 4.85 |
| Symmetry imposed | C1 | C1 | C1 |
| Map resolution FSC 0.143 (Å) | 2.48 | 2.72 | 2.9 |
| **Refinement** | | | |
| Initial model used (PDB code) | 6hre | 6hre | ModelAngelo |
| Model resolution FSC 0.5 (Å) | 2.2 | 2.6 | 3.5 |
| Map sharpening B factor (Å$^2$) | −43 | −71 | −97.8 |
| Model composition | | | |
| Non-hydrogen atoms | 3300 | 3300 | 978 |
| Protein residues | 432 | 432 | 132 |
| B factors (Å$^2$) | | | |
| Protein | 48.4 | 48.4 | 96 |
| R.m.s. deviations | | | |
| Bond lengths (Å) | 0.011 | 0.011 | 0.011 |
| Bond angles (°) | 2.009 | 1.961 | 1.935 |
| Validation | | | |
| MolProbity score | 0.9 | 0.98 | 0.5 |
| Clashscore | 0 | 0 | 0 |
| Poor rotamers (%) | 0 | 0 | 0 |
| Ramachandran plot | | | |
| Favoured (%) | 94.05 | 92.14 | 100 |
| Allowed (%) | 5.95 | 6.19 | 0 |
| Disallowed (%) | 0 | 1.67 | 0 |

Again, we observed predominantly PHFs (72%), with a minority of singlets with the Alzheimer tau fold (28%). The PHFs are identical to those found in AD brains, with a backbone root mean square deviation (r.m.s.d.) of 1.4 Å (*Figure 1E*). To further increase the proportions of PHFs-to-singlet ratio, we removed the plate from the shaker after 1 week and incubated it quiescently at 37°C for 2 more weeks. This resulted in the formation of 100% PHFs (*Figure 1—figure supplement 4*). When repeated seven times, on average, 95.3% PHFs formed, with 25% of singlets formed in a single outlier (*Figure 1—figure supplement 5*).

## PHFs of PAD12 tau are not sticky and form under various shaking conditions

While several groups have replicated our finding that tau297–391 assembles into PHFs, filament formation has required further optimisation of the assembly conditions, and the formation of other filament types has been reported (*Duan et al., 2024b*; *Glynn et al., 2024*). We previously showed that multiple filament types exist in the assembly reactions that are on-pathway to form PHFs (*Lövestam et al., 2024*). Another possible reason for the difficulties in reproducibility may be that the assembly of tau297–391 is sensitive to shaking conditions, with different structures forming when shaking is performed at 200 or at 700 rpm (*Lövestam et al., 2022*). The physical forces inside the assembly reaction vessel due to shaking are likely more difficult to control than the biochemical components of the reaction mixtures. For example, different shaking machines, or differently shaped vessels with different volumes, may all lead to different forces. The observation that the assembly of 100 µl of 0N3R-PAD12 tau inside an Eppendorf LoBind microcentrifuge tube, with orbital shaking at 500 rpm in an Eppendorf ThermoMixer C, led to the formation of PHFs with the Alzheimer fold (*Figure 1—figure supplement 3A*), suggesting that the assembly of full-length PAD12 tau is less dependent on the physics of shaking than the assembly of tau297–391.

Another disadvantage of tau297–391 PHFs is that they tend to clump together, especially after prolonged storage or freeze-thawing. Clumping of filaments complicates cryo-EM structure determination and may interfere with subsequent experiments, such as binding studies of small-molecule compounds or seeding experiments in cell culture and in animals. In contrast to tau297–391, almost all micrographs of full-length PAD12 tau showed individually dispersed filaments, even after assembled

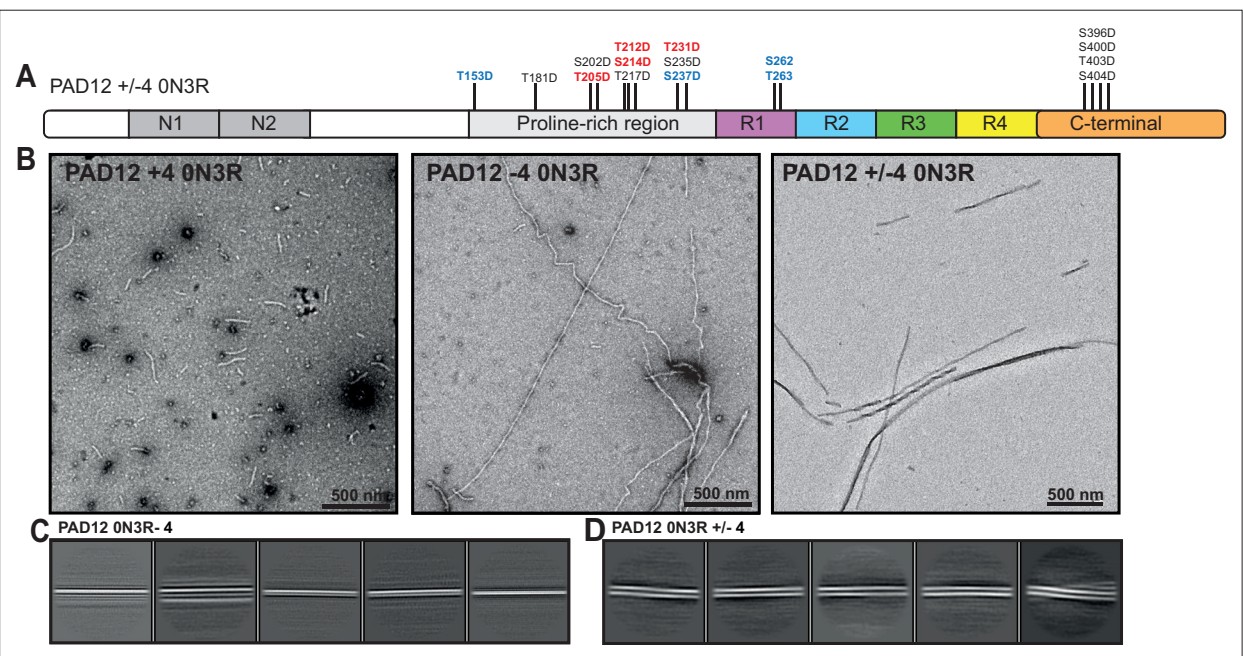

**Figure 2.** EM analysis of 0N3R PAD12+4, PAD12–4, and PAD12±4 tau constructs. (**A**) Schematic of tau sequence as in *Figure 1A*, with the four extra mutations of PAD12+4 in blue and the four mutations that were removed from PAD12–4 in red. (**B**) Negative-stain EM of filaments formed with PAD12+4 (left), PAD12–4 (middle), and PAD12±4 (right) 0N3R tau. (**C**) Electron cryo-microscopy (cryo-EM) reference-free 2D classes of filaments assembled from 0N3R PAD12–4. (**D**) Reference-free 2D classes of filaments assembled from 0N3R PAD12-4.

filaments were kept at 4°C for 2 months, or flash-frozen at –196°C and then thawed at room temperature (*Figure 1—figure supplement 3B and C*). The observation that tau 0–391 and 151–391 3R PAD12 tau constructs also did not clump together (*Figure 1—figure supplement 3D and E*) suggests that the absence of an amino-terminal fuzzy coat contributes to the stickiness of tau297–391 filaments.

## Different phosphorylation patterns lead to different filaments

We first tested whether the PAD12 mutations could also be made using glutamates instead of aspartates by substituting all 12 aspartates into glutamates. The resulting structures were again PHFs (*Figure 1—figure supplement 3A*). Next, to test whether the formation of AD PHFs depends on the specific PAD12 mutations, we tested another three 0N3R tau constructs with different mutations. In the first construct (PAD12–4), we removed four of the eight phosphomimetic mutations in the proline-rich region, leaving only T181D, S202D, T217D, and S235D. In the second construct (PAD12+4), we added four phosphomimetic mutations that also had high levels of phosphorylation by mass spectrometry of AD tau (*Wesseling et al., 2020*) (T153D, S237D, S262D, and T263D). In the third construct (PAD12±4), we removed the same four mutations as in the first construct and added the same four new mutations as in the second construct, resulting in the same net charge as in the original PAD12 construct. Assembly of the 0N3R versions of these constructs, under the same orbital shaking conditions as before, led to the formation of filaments in all three cases. However, the filaments formed with the PAD12+4, PAD12–4, or PAD12±4 constructs had morphologies that were distinct from those of PHFs; they did not twist and we were thus not able to determine their structures (*Figure 2*).

## PAD12 tau filaments can be labelled

Next, we explored the ability to label filaments of PAD12 tau using NHS-ester chemistry, which specifically targets primary amines in lysine residues. Fluorescently labelled filaments may be useful, for example, to follow seeding reactions in cells by optical microscopy. We labelled pre-assembled 0N3R-PAD12 tau filaments with DyLight-488, Alexa-647, Atto-647, Atto-565, and CF-680. We observed a fluorescent pellet after ultracentrifugation (*Figure 3C*). Cryo-EM of the resulting filaments confirmed that the PHF structure remained intact in all conditions (*Figure 3C*).

We also explored whether filaments of PAD12 tau can be biotinylated using the same NHS-ester chemistry. Biotinylation is used for protein selection and for studying protein-protein interactions by proximity labelling. Immuno-gold EM with 10 nm gold-conjugated streptavidin confirmed that pre-assembled 0N3R-PAD12 tau filaments can be extensively biotinylated (*Figure 3D*).

## In vitro seeded assembly with PAD12 tau

Next, we investigated whether AD brain-derived tau filaments could seed the assembly of PAD12 tau, and if this seeding could be sustained over more than one round. We introduced sonicated AD brain-derived tau filaments to a solution of 40 μM (~1.5 μg/μl) 0N3R-PAD12 tau and performed in vitro assembly under orbital shaking at 500 rpm (2 min on; 1 min off) for 48 hr in a buffer containing 20 mM HEPES at pH 7.3, 4 mM potassium phosphate at pH 7.2, 300 mM sodium citrate and 4 mM TCEP. Under the same conditions, but without the addition of AD seeds, we observed no filament formation within 48 hr. Although we only used extracts from 0.8 μg of AD brain tissue for reactions that contained 60 μg of recombinant (unlabelled) PAD12 tau, we observed no lag phase in Thioflavin T (ThT) fluorescence upon addition of the seeds. Instead, ThT fluorescence increased linearly directly after the seeds were added and fluorescence plateaued after 48 hr (*Figure 4A*). Cryo-EM analysis confirmed that the seeded tau filaments were predominantly (73%) PHFs (with a backbone r.m.s.d. of 1.3 Å with PDB-ID 6hre), along with a minority of single protofilaments with the Alzheimer fold (8.5%) (*Figure 4B–D*).

It was recently reported that in vitro assembly of wild-type 0N3R tau can be seeded with large amounts of AD brain-derived filaments (*Duan et al., 2024a*). The seeded filaments were shown to be PHFs, but they were not capable of seeding the assembly of wild-type 0N3R tau in a second round of seeded assembly. To test whether this is also the case for PAD12 tau, we used filaments from the first round at an estimated ratio of 1:2500 to recombinant (unlabelled) PAD12 tau for a second round of seeded assembly. We again observed a rapid increase in ThT fluorescence directly after the addition of the seeds, and cryo-EM analysis of the seeded aggregates still showed PHFs (*Figure 4A and B*), indicating that PAD12 tau can be used for multiple rounds of seeded aggregation.

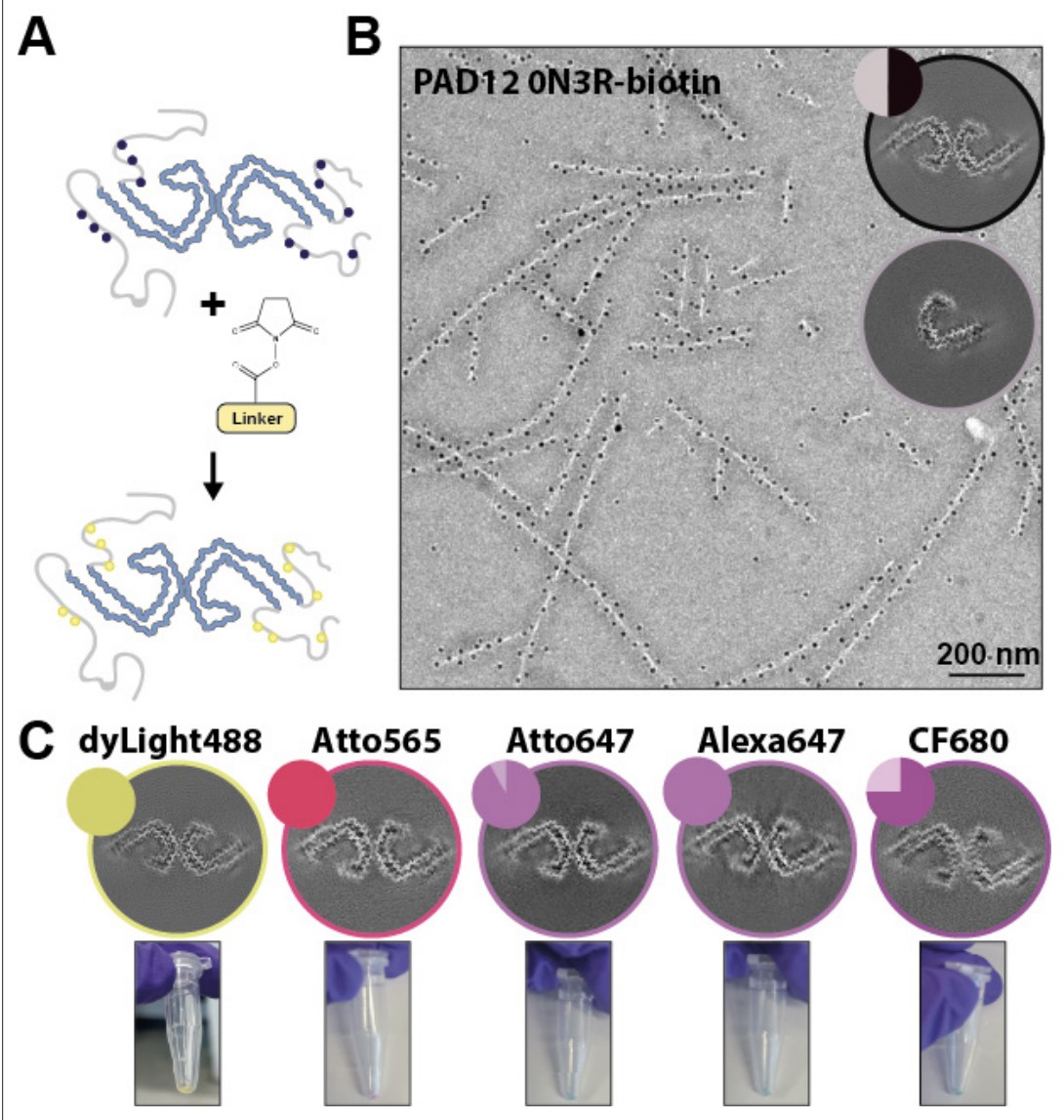

**Figure 3.** Labelling of pre-assembled PAD12 tau filaments. (**A**) Cartoon showing that filaments can be labelled via NHS-ester chemistry, which specifically targets primary amines in lysine residues. (**B**) Immuno-EM showing biotinylated tau is labelled with streptavidin-coated 10 nm gold particles. (**C**) Electron cryo-microscopy (cryo-EM) reconstructions of PAD12 3R tau (top) labelled with different fluorophores. The circular inset shows a cross-section of the corresponding cryo-EM reconstruction, and the pie charts show the distributions of the different filament types (paired helical filaments [PHFs] in solid colours; singlets in lighter colours). The coloured pellets (bottom) indicate successful labelling of the filaments.

## Seeding of tau reporter cells with PAD12 PHFs

We then used a biosensor cell line that over-expresses hemagglutinin (HA-)tagged tau297–391 in HEK293T cells (see Materials and methods) and compared its seeding response to increasing amounts of PHFs made with tau297–391, PAD12 0N3R, or PAD12 0N3R:0N4R (*Figure 5A and B*). These cells also yielded robust seeding, in a concentration-dependent manner, when exposed to seeds extracted from AD brains (*Figure 5—figure supplement 1A*). Comparing the recombinant PHFs of tau297–391 and PAD12 0N3R or PAD12 0N3R:0N4R tau, we observed efficient seeding at low concentrations (~5 ng) of PAD12 PHFs, whereas tau297–391 did not lead to seeding at the same concentrations. At higher concentrations, seeding with tau297–391 PHFs was less efficient and showed greater variability among replicates than seeding with PAD12 PHFs. Using appropriate excitation and emission settings to visualise seeds of PAD12 0N3R filaments that were labelled with DyLight-488 inside the biosensor

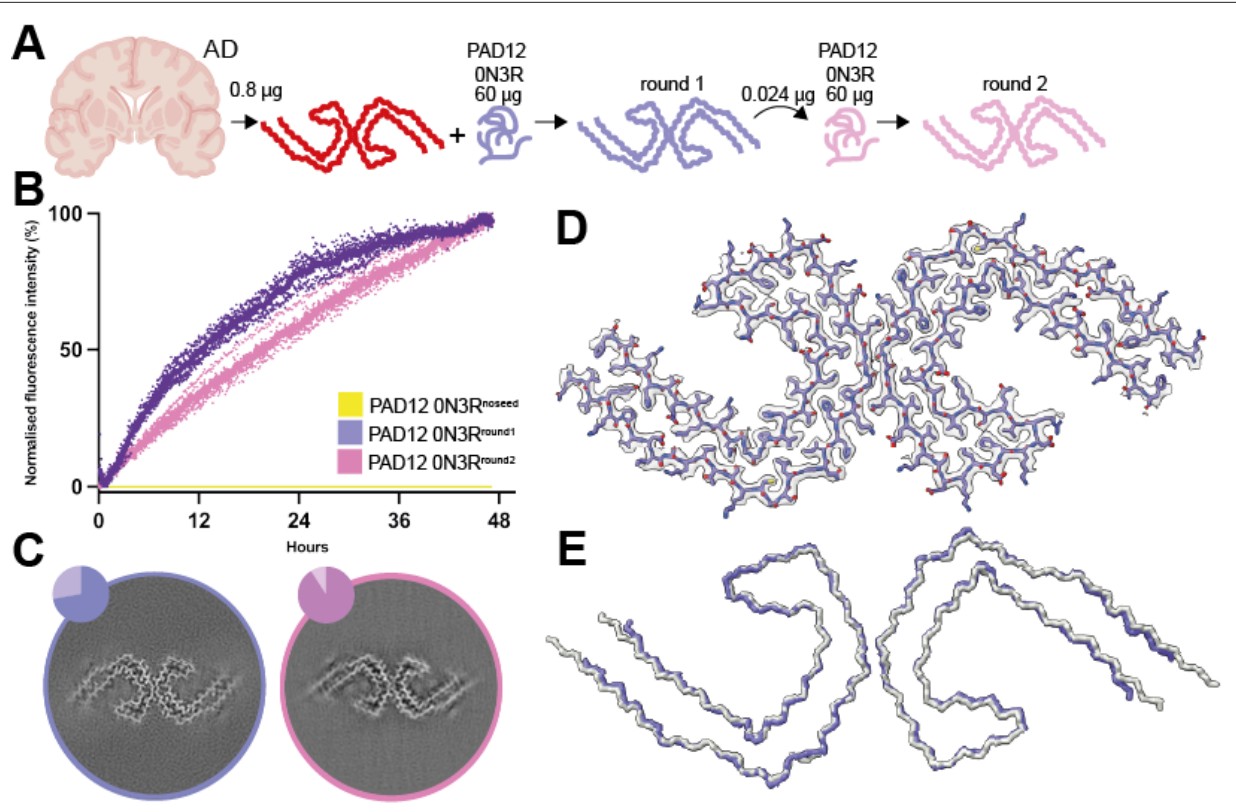

**Figure 4.** In vitro seeded assembly with PAD12 tau. (**A**) Schematic of experimental approach for the seeded assembly of PAD12 0N3R. Small amounts of brain material are used to seed the assembly of PAD12 0N3R (round 1). The filaments formed from round one are used as seeds for a second round. (**B**) Thioflavin T (ThT) fluorescence profile of the Alzheimer's disease (AD)-seeded (purple), second-round seeding (pink), and non-seeded control (yellow) N=9. The circles are individual measurements (normalised for each reaction). (**C**) Cross-sections of electron cryo-microscopy (cryo-EM) reconstructions perpendicular to the helical axis, with a thickness of approximately 4.7 Å and pie charts showing the distribution of filament types for the first (left) and second (right) round of seeding (pink/purple paired helical filaments [PHFs], yellow single protofilaments with the Alzheimer fold (not shown); grey discarded filaments). (**D**) Cryo-EM density map of AD-seeded 0N3R PAD12 tau (transparent grey) with the superimposed fitted atomic model. (**E**) Main chain trace of in vitro seeded PHF (purple) overlaid with AD PHF (grey; PDB-ID: 6hre).

cells, we observed colocalisation of the original seeds with the HA-297–391 puncta (*Figure 5C*; *Figure 5—figure supplement 1B and C*).

## Phosphomimetic mutations in the carboxy-terminal domain affect the FIA region

We used solution-state NMR to explore the effects of PAD12 mutations on the tau monomer. Because of size limitations imposed by this technique, we used two truncated tau constructs. The first construct comprised residues 151–391 of 4R tau, with or without the eight phosphomimetic mutations of PAD12 in the proline-rich region. The second construct comprised residues 297–441 of 4R tau, with or without the four phosphomimetic mutations of PAD12 in the carboxy-terminal domain.

The effects of the phosphomimetic mutations in the proline-rich region of the tau151–391 construct were subtle. Chemical shift perturbations (CSPs) between wild-type and PAD12 tau151–391 were largest around residues 198–222, which harboured mutations at S202, T205, S212, S214, and T217, and around residues 237–244, which were adjacent to the mutations at T231 and S235 (*Figure 6—figure supplement 1A*; *Figure 6—figure supplement 2*). Analysis of the backbone Cα and Cβ chemical shifts (which are sensitive to backbone torsion angles) revealed little difference in secondary structure propensity in the 297–391 region between wild-type tau151–391, PAD12 tau151–391, and tau297–391 (*Figure 6—figure supplement 1B*), or in the proline-rich region of wild-type and PAD12 tau151–391 (although its 26 prolines may prevent any potential alterations to secondary structure). Heteronuclear NOE (HetNOE) analysis (*Figure 6—figure supplement 1C*), which reports on picosecond timescale

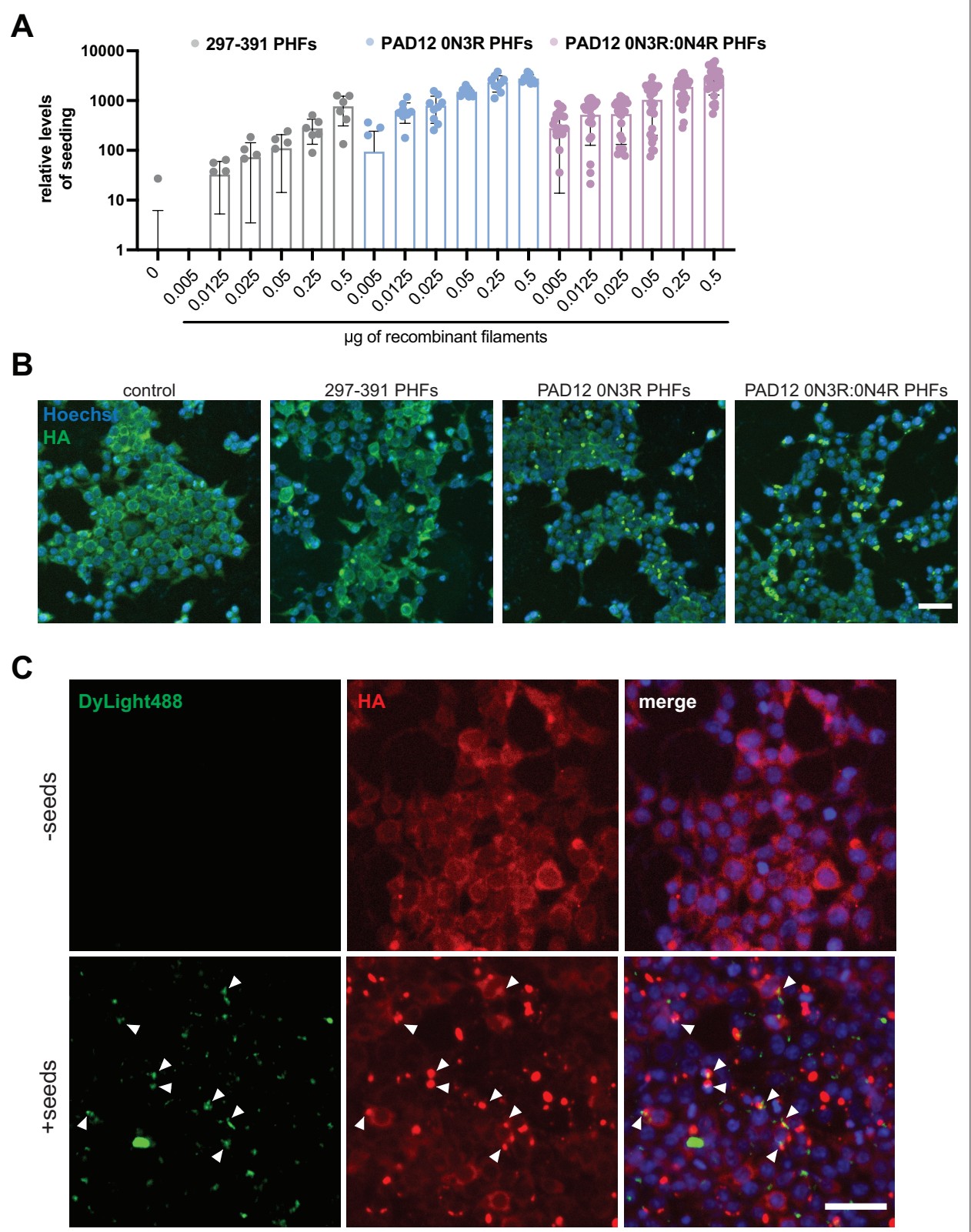

**Figure 5.** Cellular seeding with recombinant tau filaments. (**A**) Box plot showing the number of detected seeds, which were normalised to the number of cells and compared to mock-treated control cells (n≥10,000 cells/condition analysed). Graph represents mean values; error bars represent standard deviation. (**B**) Images from control (-seed) and cells seeded with 0.25 µg of assembled tau297–391 paired helical filaments (PHFs), PAD12 0N3R PHFs and PAD12 0N3R:0N4R PHFs. Fixed cells were stained against hemagglutinin (HA) for labelling over-expressed tau297–391 (green) and Hoechst (blue)

*Figure 5 continued on next page*

*Figure 5 continued*

for labelling of the nucleus. Scale bar, 50 µm. (**C**) Images from control cells without the addition of seeds (-seed) and cells seeded with 0.25 µg of PAD12 0N3R PHFs that were pre-labelled with DyLight-488. Fixed cells were stained against HA for labelling over-expressed tau (red) and Hoechst (blue) for labelling nuclei. Scale bar, 50 µm.

The online version of this article includes the following figure supplement(s) for figure 5:

**Figure supplement 1.** Cellular seeding with Alzheimer's disease (AD) brain-derived tau and with PAD12 0N3R tau filaments labelled with DyLight-488.

backbone mobility, revealed increased rigidity of PAD12 tau151–391 compared to wild-type tau151–391, predominately around the mutation sites. Relative peak intensities (*Figure 6—figure supplement 1D*) showed some differences in the proline-rich region, where peak attenuation indicated greater conformational sampling for wild-type tau151–391 than for PAD12 tau151–391. However, the subtle differences between these two constructs do not suggest an obvious reason for the increased propensity of the PAD12 tau151–391 to form filaments.

By contrast, comparison of wild-type and PAD12 tau297–441 revealed interesting differences. Secondary chemical shift analyses (*Figure 6—figure supplement 3A*) suggested that residues 393–404, which harbour the four mutation sites, had an increased propensity to adopt extended conformations in the PAD12 construct, possibly because of electrostatic repulsion between the negatively charged aspartates. Residues 405–407 exhibited a strong preference for helical backbone torsion angles, indicating the presence of a turn-like structure, in the PAD12 construct, but not in wild-type tau297–441. The effect of the phosphomimetic mutations on these residues was also echoed in the CSPs (*Figure 6—figure supplement 3B*; *Figure 6—figure supplement 4*). HetNOE analysis suggests that the PAD12 construct is more rigid on the picosecond timescale than wild-type tau297–441 between residues 305–311 and 398–412 (*Figure 6—figure supplement 3C*). Finally, peak intensity analysis revealed that residues 297–319 of wild-type tau297–441 displayed reduced intensities compared to tau297–391 and PAD12 tau297–441 (*Figure 6—figure supplement 3D*). Residues 392–404 of wild-type tau297–441 also showed similar attenuated peak intensities compared to PAD12 tau297–441.

Residues 302–316 form the ordered core of the FIA in the assembly of tau297–391 into PHFs or CTE filaments. We previously hypothesised that partial rigidity of these residues in tau297–391 monomers may reduce the entropic cost of nucleating new filaments (*Lövestam et al., 2024*). The observation that, by HetNOE analysis, the same region appears more rigid in PAD12 tau297–441 than in wild-type tau297–441 may contribute to the ability of the former to assemble into filaments. In addition, it could be that the reduced peak intensities for residues 297–319 and 392–404 stem from conformational exchange broadening as a result of a transient intramolecular interaction between their respective IVYK motifs in repeat 3 (residues 308–311) and in the carboxy-terminal region (residues 392–395). Such a hairpin-like interaction, which may resemble that between the two protofilaments of the FIA, could inhibit the filament formation in the absence of phosphomimetic mutations. A similar interaction, which was also impeded by phosphomimetic mutations in the carboxy-terminal domain (S396E and S404E), has been hypothesised to occur between the microtubule-binding repeat region of tau and its carboxy-terminal domain based on FRET measurements (*Jeganathan et al., 2008*). Earlier, phosphorylation of tau had been observed to lead to a pronounced change in electrophoretic mobility, which was suggested to reflect a conformational change (*Lindwall and Cole, 1984*).

To investigate this further, we also tested a tau construct comprising residues tau297–441 without the phosphomimetic mutations, but with a deletion of residues $_{392}$IVYK$_{395}$ (Δ392–395). Filaments formed rapidly and the cryo-EM structure showed that the ordered core consisted of the amino-terminal part of the construct spanning residues 297–318 (*Figure 6B*). NMR analysis (*Figure 6—figure supplement 5B*) showed that the tau297–441 Δ392–395 construct exhibited similar backbone rigidity properties to the tau297–441 PAD12 construct, despite peak locations and local secondary structural propensities being more similar to the wild-type tau297–441 (*Figure 6—figure supplement 5A*; *Figure 6—figure supplement 6*). HSQC peak intensities in the 297–319 and 392–404 regions of tau297–441 Δ392–395 (*Figure 6A*, expanded from *Figure 6—figure supplement 5C*) were like those in the tau297–441 PAD12. These data suggest that the IVYK deletion has a similar effect as the phosphomimetics on residues 396, 400, 403, and 404 on disrupting an intra-molecular interaction between the FIA core region and the carboxy-terminal domain, which may therefore be mediated by interactions between the two IVYK motifs that are similar to those observed in the FIA (*Lövestam et al., 2024*).

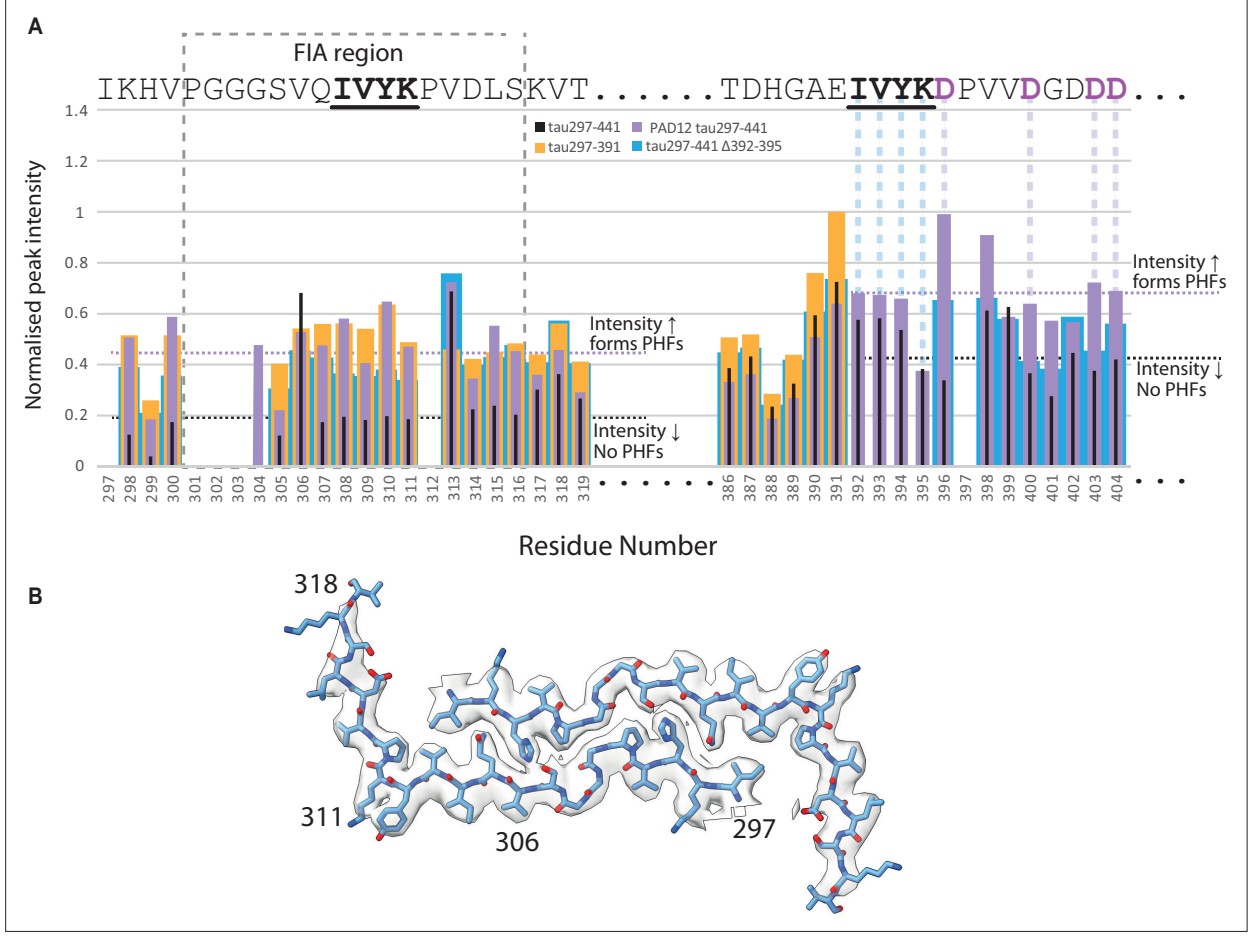

**Figure 6.** Nuclear magnetic resonance (NMR) and electron cryo-microscopy (cryo-EM) of C-terminal tau297–441, tau297–441 PAD12, and tau297–441 Δ392–395. (**A**) Peak height differences for selected residues in heteronuclear single quantum coherence (HSQC) spectra as shown by normalised peak intensity. Values for tau297–391 are shown in gold, tau297–441 in black, PAD12 tau297–441 in lilac, and tau297–441 Δ392–395 in blue. The grey dashed line box indicates the first intermediate amyloid (FIA) region (residues 301–316). Lilac dashed lines are PAD12 mutations and blue dashed lines are the residues deleted in the Δ392–395 tau construct. Horizontal dotted lines highlight the differences in peak intensity between tau297–441 (black, does not form paired helical filaments [PHFs]) and PAD12 tau297–441 (purple, does form PHFs) in the FIA region and around the PAD12 mutation site. Both tau297–391 (gold) and tau297–441 Δ392–395 (blue) have peak intensity values like those of PAD12 tau297–441. Full residue information is shown in *Figure 6—figure supplement 5C*. (**B**) The cryo-EM structure of tau297–441 Δ392–395. Cryo-EM density map in transparent grey with the superimposed fitted atomic model in blue. All filaments were of the same type.

The online version of this article includes the following figure supplement(s) for figure 6:

**Figure supplement 1.** Nuclear magnetic resonance (NMR) spectroscopy of wild-type and PAD12 tau151–391.

**Figure supplement 2.** Heteronuclear single quantum coherence (HSQC) peak assignment of wild-type and PAD12 tau151–391.

**Figure supplement 3.** Nuclear magnetic resonance (NMR) spectroscopy of wild-type and PAD12 tau297–441.

**Figure supplement 4.** Heteronuclear single quantum coherence (HSQC) peak assignment of wild-type and PAD12 tau297–441.

**Figure supplement 5.** Nuclear magnetic resonance (NMR) spectroscopy of tau297–441 Δ392–395.

**Figure supplement 6.** Heteronuclear single quantum coherence (HSQC) peak assignment of tau297–441 Δ392–395.

## Discussion

Tau filaments that are extracted from the brains of individuals with AD using sarkosyl contain full-length, hyperphosphorylated tau of all six isoforms (*Goedert et al., 1992*). By contrast, the assembly of full-length recombinant tau into filaments requires the presence of negatively charged cofactors. We previously showed that heparin-induced in vitro assembly of full-length recombinant tau yields structures that are different from those of filaments extracted from AD brains (*Zhang et al., 2019*), but that truncated tau297–391 can be assembled into PHFs in the absence of cofactor (*Lövestam et al.,*

*2022*). We also reported that the introduction of four phosphomimetic mutations in the carboxy-terminal domain of tau297–441 yields individual protofilaments with the Alzheimer fold. In this paper, we show that an additional eight phosphomimetic mutations in the proline-rich region are sufficient for the assembly of recombinant full-length tau into PHFs. These findings show for the first time the nucleation-dependent assembly of full-length recombinant tau into PHFs.

The proline-rich region of unmodified tau is positively charged. Our findings with the PAD12+4, PAD12–4, and PAD12±4 constructs show that the assembly of full-length tau into PHFs is not merely dependent on the introduction of negative charges in this region: which residues are mutated matters too. This is in agreement with studies by others using pseudo-phosphorylation of recombinant tau and phosphorylation of recombinant tau by some protein kinases (*Haase et al., 2004*; *Lee et al., 2001*). We cannot exclude the possibility that each construct requires its own optimisation of assembly conditions and that constructs with modifications other than those of PAD12 could be made to form PHFs too. Previous studies have reported that phosphorylation or phosphomimetic mutations of different subsets of the PAD12 mutations could lead to the assembly of tau in vitro and in cells (*Alonso et al., 2010*; *Despres et al., 2017*). However, for none of these studies are cryo-EM structures available to show that the resulting filaments were PHFs. The definition of a minimal set of phosphomimetic mutations that is necessary for the spontaneous assembly of full-length tau into PHFs would require the systematic removal of PAD12 mutations, each with their own optimisation of assembly conditions and cryo-EM structure determination to verify that the correct structures are formed. While such work may yield further insights in the future, we feel that the assembly of PAD12 full-length tau into PHFs warrants dissemination before such explorations are performed.

Like tau filaments in disease, and unlike filaments of tau297–391, PAD12 tau filaments have a fuzzy coat. The role of tau's fuzzy coat in the molecular mechanisms of disease remains poorly understood. For example, AD-seeded PAD12 tau filaments can be used as seeds in a second round of in vitro seeded assembly, whereas AD-seeded unmodified tau filaments were reported to be seeding-incompetent (*Duan et al., 2024a*), even though both tau constructs formed PHFs in the first round of seeding. It is also possible that the fuzzy coat affects the interactions between tau filaments and cellular components, such as extracellular receptors for the uptake of tau filaments (*De Rocque et al., 2021*), or macromolecular complexes that degrade them (*Koopman et al., 2022*). Moreover, the fuzzy coat may interfere with the binding of candidate ligands for positron emission tomography or biologics designed to target the ordered cores of tau filaments in disease. Further research to explore the roles of the fuzzy coat and its post-translational modifications is required, and the ability to make PHFs with a fuzzy coat from recombinant PAD12 tau will facilitate this.

Hyperphosphorylation of tau disrupts its ability to interact with microtubules and has been implicated in filament assembly (*Iqbal et al., 2016*). Phosphorylation of the fuzzy coat may facilitate spontaneous filament assembly by reducing the entropic cost of nucleating new filaments. In addition, the negative charges of hyperphosphorylated tau in filaments may actively recruit soluble tau monomers with less phosphorylation through electrostatic attraction. This could explain why the assembly of wild-type full-length tau with large amounts of AD brain-derived seeds worked for a single round of seeding, with the resulting filaments being unable to seed the assembly of wild-type full-length tau in a second round (*Duan et al., 2024a*). It is striking that phosphorylation patterns of tau from AD brains exclude residues from the ordered core of the PHF. In agreement, no cryo-EM density for phosphorylation groups is visible in the ordered cores of any of the reported tau filaments from postmortem brains (*Scheres et al., 2023*). Because the ordered cores of PHFs comprise the third and fourth microtubule-binding domains of tau, it is possible that these residues are protected from phosphorylation, while tau is bound to microtubules. In fact, tau molecules with phosphorylation of the residues in the ordered core of PHFs may not be able to form filaments, due to the size of the phosphate groups and the repulsive forces between their charge. Phosphorylation of tau in the fuzzy coat is a physiological mechanism that regulates its binding to microtubules (*Iqbal et al., 2016*).

Our NMR data provide insights into the mechanism by which phosphorylation in the fuzzy coat of tau, or truncations of tau, lead to the assembly of filaments with ordered cores of residues that are themselves not phosphorylated. HSQC peak intensity differences between unmodified tau 297–441, PAD12 tau 297–441, and tau297–391 suggest that phosphorylation of the fuzzy coat, particularly near the $_{392}$IVYK$_{395}$ motif in the carboxy-terminal domain, affects the conformation of the residues of tau that become ordered in the FIA (*Lövestam et al., 2024*). Removal of residues $_{392}$IVYK$_{395}$ in the

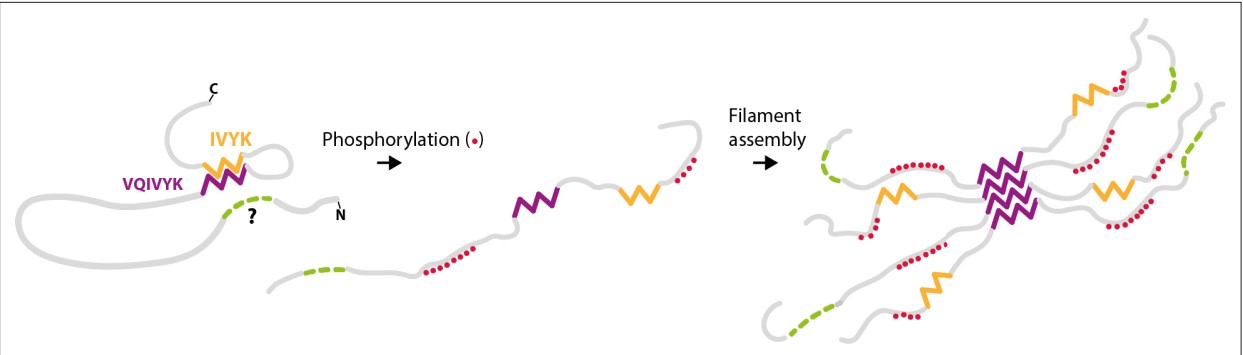

**Figure 7.** Hairpin model of phosphorylated tau filament assembly. In unmodified tau monomer, residues $_{392}$IVYK$_{395}$ (orange) in the carboxy-terminal domain interact with residues $_{306}$VQIVYK$_{311}$ (purple) in the core-forming region of tau. A similar interaction between the amino-terminal domain of tau (green) may also exist, but the residues involved in this interaction are unknown. Upon phosphorylation (red) of residues in the amino-terminal and the carboxy-terminal domains of tau, these interactions are disrupted, which then leads to filament nucleation through inter-molecular interactions of residues $_{306}$VQIVYK$_{311}$ in the first intermediate amyloid (FIA).

carboxy-terminal domain of tau 297–441 led to rapid filament formation in the absence of phosphomimetics, while HSQC peak intensity differences for this construct indicate similar backbone rigidity compared to tau 297–441 without the deletion, but with the four PAD12 mutations in the carboxy-terminal domain. Combined, these observations support a model where the $_{392}$IVYK$_{395}$ motif in unmodified full-length tau monomers interacts with the $_{308}$IVYK$_{311}$ motif, thus inhibiting filament formation by preventing the formation of the nucleating species, the FIA. Phosphorylation of nearby residues 396, 400, 403, and 404, or truncation at residue 391, disrupts this interaction and leads to filament formation. This model agrees with the previously proposed paperclip model of tau (*Jeganathan et al., 2008*), although the corresponding interaction between the amino-terminal domain of tau and the core-forming region remains unknown (*Figure 7*). The observation that PAD12 tau migrates at a higher apparent molecular weight in SDS-PAGE than predicted (*Figure 1—figure supplement 1*) also supports a model in which the PAD12 mutations lead to a more extended conformation. It is possible that phosphorylation of tau in its normal physiological role of regulating microtubule stability (*Iqbal et al., 2016*) does not extend to all the residues that are required for filament assembly, but this may sometimes be the case. It remains to be seen if tau filaments form occasionally as a result of this physiological regulation, followed by their degradation, or if tau filament formation is always the consequence of a pathological event.

It could be that distinct post-translational modification patterns are important for the assembly of tau into protofilament folds that are specific for the other tauopathies. To explore this, similar approaches as described here for AD could be applied to other tauopathies, provided proteomics data on tau filaments from these diseases becomes available. In the meantime, the present findings suggest that hyperphosphorylation of tau is sufficient for the formation of the Alzheimer fold. It remains to be seen if other post-translational modifications of tau can also give rise to PHFs. The ability to form PHFs from recombinant PAD12 tau will enable further research into the molecular mechanisms of tau aggregation and its role in AD. The observations that PAD12 tau filaments can be labelled, and that individual filaments are stable in solution for weeks, will facilitate their use in the seeding of tau in cells and in animals, as well as for high-throughput binding assays with potential therapeutic or diagnostic compounds, and their structure determination. Understanding how and why tau adopts specific folds in the different tauopathies, and whether or how these folds affect the different diseases, may provide new avenues for therapeutic development.

## Materials and methods
### Protein purification
Tau constructs in pRK172 containing 0N4R or 0N3R cDNA tau were made using in vivo assembly (*García-Nafría et al., 2016*) and transformed into BL21-CodonPlus (DE3)-RIPL competent cells (Agilent) for expression. Transformed cells from one plate were resuspended in 2xTY with 5 mM

magnesium chloride and 100 mg/l ampicillin; inoculated in 4 l of 2xTY with 5 mM magnesium chloride and 100 mg/l of ampicillin; grown to an optical density of 0.8 at 600 nm; and induced with 0.6 mM isopropyl β-D-1-thiogalactopyranoside (IPTG) for 3 hr at 37°C. For $^{15}$N and $^{13}$C-labelled tau, bacteria were grown in isotope-enriched M9 minimal medium containing 1 g/l of [$^{15}$N]ammonium chloride and 2 g/l of [$^{13}$C]glucose (Sigma), supplemented with 1.7 g/l yeast nitrogen base (Sigma). Expression was induced with 0.8 mM IPTG at 18°C overnight. Cells were harvested by centrifugation for 35 min at 4°C 4400×$g$. Pelleted cells were flash-frozen in liquid nitrogen and snapped into a beaker with a stirring bar. Per 6 l culture, 50–60 ml of Buffer A (50 mM MES at pH 5.5–6.5, with 250 mM sodium chloride, 2 mM ethylenediaminetetraacetic acid [EDTA], 5 mM magnesium chloride, 10 mM dithiothreitol [DTT], 0.03 mM chymostatin, 0.1 mM phenylmethylsulphonyl fluoride [PMSF], 0.1 mM 4-(2-aminoethyl) benzenesulfonyl fluoride hydrochloride [AEBSF], supplemented with cOmplete EDTA-free Protease Inhibitor Cocktail [Roche] [three tablets per 1 l and an additional tablet in 50 ml], 40 µg/mL DNAse I [Sigma] and 10 µg/mL bovine pancreas RNAse [Sigma]) were added. The pH of the buffer was adjusted to be one unit lower than the isoelectric point (pI) of each tau construct used. Cells were lysed by sonication (5 s on; 10 s off at 40% amplitude for 4 min in a Sonics VCX-750 Vibra Cell Ultra Sonic Processor) followed by 20 s at 90% amplitude. Lysed cells were further supplemented with 40 µg/ml DNAse (Sigma) and 10 µg/ml RNAse (Sigma) and left to stir for 10 min at room temperature. Cell lysates were centrifuged at 20,000×$g$ for 35 min at 4°C. The lysate was diluted fivefold, to get a sodium chloride concentration of 50 mM. The supernatant was loaded onto a HiTrap CaptoS column (GE Healthcare) and eluted with a 1 M sodium chloride gradient (Buffer A+1 M sodium chloride). Fractions (3.5 ml) were analysed by SDS-PAGE and protein-containing fractions were precipitated using 0.38 g/ml of ammonium sulphate at 4°C for 45 min. Precipitated protein was pelleted at 20,000×$g$ for 30 min at 4°C, and resuspended in 10 mM potassium phosphate buffer at pH 7.2, with 10 mM DTT, and loaded onto a Superdex 75 pg 16/600 size exclusion column (GE Healthcare) using a flow rate of 1 ml/min. Protein-containing fractions were pooled and concentrated using a vivaspin 3 kDa concentrator (Merck) until protein concentrations reached 0.2–1 mM.

## In vitro assembly

Protein samples were thawed on ice and filtered (Costar Spin-X Centrifuge Tube Filters, 0.22 µm). The membranes of the filters were washed with 20 µl water before applying the sample. Protein sample concentrations after filtering were measured using a NanoDrop spectrophotometer. Assembly reactions were prepared in Eppendorf Protein LoBind tubes. All buffers were filtered. Reactions were prepared by subsequently mixing water, buffering agent (from a 1 M HEPES stock at pH 7.28, or a 1 M phosphate buffer stock at pH 7.2), TCEP (from a 100 mM stock), salt (from a 5 M sodium chloride, 1 M potassium chloride, 1 M potassium citrate, 1 M sodium citrate, 1 M sodium malate) or ATP (from a 750 mM stock), protein, and lastly ThT (from a 150 µM stock). Wells in a 384-well plate were flushed with 100 µl water prior to setting up the reaction which were prepared in batch. Aliquots of 30–40 µl were dispensed in each well, making sure no bubbles were present. Each reaction had an empty well next to it to prevent cross-contamination by evaporation. Shaking conditions were 2 min on, 1 min off, at 500 rpm orbital shaking at 37°C, reading ThT fluorescence every 10 min.

## Seeded assembly in vitro

Frozen frontal cortex from the case of AD used in *Fitzpatrick et al., 2017*, was thawed at room temperature, and 100 mg tissue was homogenised in 10 vol of extraction buffer (20 mM Tris, pH 7.4, 5 mM EGTA, 5 mM EDTA, 800 mM sodium chloride, 10% sucrose, and 1% sarkosyl) and incubated at 37°C for 30 min with shaking. The samples were centrifuged at 20,000×$g$ for 15 min at room temperature, followed by ultracentrifugation of the supernatants at 150,000×$g$ for 30 min. The pellets were resuspended in 100 µl of extraction buffer and incubated at 37°C for 3 hr with shaking at 500 rpm (Eppendorf ThermoMixer C). The samples were then centrifuged at 20,000×$g$ for 20 min, and the supernatants ultracentrifuged at 150,000×$g$ for 30 min at room temperature. Pellets were resuspended in 50 µl of 20 mM Tris at pH 7.4, with 100 mM sodium chloride and stored at 4°C. One µl was diluted in 99 µl of buffer 20 mM HEPES pH 7.28, and sonicated using a UP200St with VialTweeter operating at 200 W. This served as a ×100 stock solution of seeds (at ~2 µg of original brain tissue per µl of stock solution). A total of 0.4 µl of the AD stock was used per seeding reaction in a 40 µl reaction, corresponding to a total amount of 0.02 µg of brain tissue per µl of the seeded assembly reactions.

Assembly reactions were prepared as described above, with the addition of seeds preceding that of ThT.

## Lentivirus-mediated generation of tau biosensor cells

HEK293T cells were purchased from ATCC (Catalogue number [Cat#]: CRL-3216; RRID:CVCL_0063) and were maintained in High Glucose GlutaMAX Pyruvate Dulbecco's modified Eagle medium (DMEM) (Thermo Fisher Scientific; Cat# 31966047) supplemented with 10% foetal bovine serum (FBS) (Thermo Fisher Scientific; Cat# 10270106), 100 U/ml penicillin, 100 µg/ml streptomycin, and grown at 37°C in 95% $O_2$/5% $CO_2$. HEK293T cells stably expressing tau297–391 with an N-terminal human influenza hemagglutinin (HA-) tag were generated as described (*Elegheert et al., 2018*). Briefly, the construct was amplified using the following pair of primers 4R-HA_I297_EcoRI_fwd: CTGACTGACTGAGAAT TCgccaccATGTACCCATACGATGTTCCAGATTACGCTATCAAACACGTCCCGGGAGGC and 4R-E391_ XhoI_rev: CTGACTGACTGActcgagTctactaCTCCGCCCCGTGGTCTGTC. The resulting PCR products were cloned into the lentiviral vector pHR_SFFV (Addgene; Cat# 79121; RRID:Addgene_79121) using the restriction enzymes EcoRI and XhoI (New England Biolabs). Lentiviral particles were produced using the packaging plasmid psPAX2 (Addgene; Cat# 12260; RRID:Addgene_12260) and the VSV-G envelope expressing plasmid pMD2.G (Addgene; Cat# 12259; RRID:Addgene_12259). All three plasmids were co-transfected in a 1:1:1 ratio into empty HEK293T cells using Lipofectamine 3000 (Thermo Fisher Scientific; Cat# L3000015) according to the manufacturer's instructions. After 3 days, cell culture supernatants containing lentiviral particles were collected and passed through a 0.45 µm filter. The clarified supernatants were diluted by the addition of 25% vol fresh medium and used to transduce HEK293T cells with Polybrene (Merck; Cat# TR-1003-G) at a final concentration of 10 µg/ml for 72 hr. The HEK293T cells used in this study tested negative for mycoplasma contamination and their identity was confirmed as HEK293T by short tandem repeat profiling (Eurofins Genomics, CLA service), consistent with established reference profiles.

## Cell seeding assays

Before use, recombinant filaments were sonicated in a water bath (QSonica) for 15 s at 50% amplitude; brain-derived seeds were sonicated during the sarkosyl-extraction procedure. Brain-derived seeds were prepared from 1 to 2 g of frontal cortex, which was homogenised in 15 vol of extraction buffer. The samples were then centrifuged at 10,000×$g$ for 10 min at 4°C and the supernatants passed through a 70 µm cell strainer. The clarified homogenates were centrifuged at 150,000×$g$ for 1 hr at 4°C, and the pellets resuspended in 700 µl extraction buffer per g tissue, followed by sonication with a Microson XI-2000 Ultrasonic Cell Disruptor (Misonix) for 20 s and centrifugation at 10,000×$g$ for 10 min. The supernatants were diluted threefold in 50 mM Tris-HCl, pH 7.4, containing 150 mM NaCl, 10% sucrose, and 0.2% sarkosyl, followed by centrifugation at 150,000×$g$ for 1 hr. The final pellets were resuspended in 50 µl/g tissue of 20 mM Tris-HCl, pH 7.4, 100 mM NaCl.

Cell seeding experiments were performed as described, with minor modifications (*McEwan et al., 2017*). Approximately 15,000 cells were plated in black 96-well plates that were pre-coated with poly-D-lysine (Merck; Cat# A-003-E, final coating concentration of 50 µg/ml) and left to adhere overnight. The next day, cells were rinsed with PBS and added to 100 µl Opti-MEM medium (Thermo Fisher Scientific; Cat# 31985062) containing the indicated amounts of recombinant or brain-derived assemblies in complex with 1 µl of Lipofectamine 2000 (Thermo Fisher Scientific; Cat# 11668019). Cells were incubated at 37°C for 1 hr, and the Lipofectamine-mediated delivery of the assemblies was stopped by the addition of 100 µl DMEM containing 10% FBS. Two days after the addition of seeds, the cells were fixed with cold methanol for 3 min at room temperature and incubated overnight at 4°C with an anti-HA antibody (diluted 1:2000) (BioLegend; Cat# 901502; RRID:AB_10064068). Cells were then rinsed three times with PBS and incubated for 1 hr at room temperature with an Alexa647-conjugated goat anti-mouse antibody (diluted 1:1000) (Thermo Fisher Scientific; Cat# A-21235; RRID:AB_2535804). After rinsing three times with PBS, cell nuclei were stained with 1 µg/ml Hoechst dye (Thermo Fisher Scientific; Cat# H3570) for 10 min and images were acquired at 405 and 647 nm on a Ti2-E High Content Microscope (Nikon) using the 10× objective. For the experiments with the Dy-Light488-labelled assemblies, images were acquired at 488 nm. Nine fields per well were read in a horizontal serpentine acquisition mode with a 10× objective, and the downstream analysis was performed using the Fiji software (*Schindelin et al., 2012*). For nuclear counting, the images acquired

at 405 nm were locally subtracted for background using the Rolling ball algorithm, and cells were segmented based on nuclear staining using the Median filter and Find Maxima tools, with the option of 'Segmented Particles above lower threshold' activated. Seeded aggregates were detected in the 488 nm images and quantified using the ComDet plugin (*Katrukha, 2020*) in Fiji. Positive puncta were determined by an approximate particle size of 6 pixels, while the intensity threshold was variable between experiments and was based on detecting the minimum number of aggregates in the unseeded condition. Finally, the relative levels of seeding were calculated as the number of aggregates in each field was normalised to the corresponding number of cells and was then compared to the untreated control.

## Labelling of pre-assembled filaments

Filaments were labelled as described in the provider protocol description. A 50 µg aliquot of DyLight 488 NHS Ester (Thermo Scientific) was dissolved in 20 µl HEPES buffer (pH 7.28), followed by the addition of 50 µl of filament solution. The reactions were incubated for 1 hr at room temperature in the dark. For the red colours, stocks were made in DMSO at 10 mg/ml and aliquoted (5 µl each). One aliquot was thawed and 1 µl was incubated with 50 µl of tau filaments. For biotinylation, 1 mg of EZ-Link Sulfo-NHS-Biotin (Thermo Scientific) was dissolved in 300 µl dimethylsulfoxide (DMSO). Then, 0.5 µl of the biotin solution was added to 20 µl of filament solution, maintaining a 5:1 (biotin:tau) molar ratio. The reaction was incubated for 30 min at room temperature. Subsequently, reactions were ultracentrifuged at 100,000×*g* for 20 min at room temperature. Pellets were resuspended in 20 mM HEPES buffer (pH 7.28) with 100 mM potassium citrate, using 50 µl and 20 µl for the DyLight-labelled filaments and the biotinylated filaments, respectively. To confirm successful labelling, 3.5 µg of DyLight-labelled filaments were analysed via SDS-PAGE (4–20%), followed by exposure at a wavelength of 488 nm. Biotinylation was validated by immuno-EM.

## Negative-stain EM and immuno-EM

For negative-stain EM, samples were diluted 10-fold (to ~4 µM) in 20 mM HEPES at pH 7.3, and applied to glow-discharged carbon grids for 1 min, blotted, washed with 4 µl water, blotted and stained with 2% uranyl acetate for 2 min, blotted and imaged.

For immuno-EM, after applying filaments to the grid for 2 min, they were blotted and blocked with 0.5% fish gelatin in PBS for 5 min, washed with 200 µl of water, and incubated with 50 µl streptavidin coated with 10 nm gold nanobeads (Sigma) in a ratio of 1:20 in blocking buffer, incubated for 30 min, further washed with 200 µl of water, blotted and stained with 2% uranyl acetate for 2 min, before being blotted and imaged by transmission electron microscopy at room temperature.

## Cryo-EM data acquisition

Protein sample aliquots of 3 µl were applied to glow-discharged holey carbon grids (Quantifoil Au R1.2/R1.3 300 mesh), blotted with filter paper, and plunge-frozen into liquid ethane using an FEI Vitrobot Mark IV (100% humidity, 4°C). The cryo-EM images in *Figure 1—figure supplement 3* were recorded on TFS Glacios using a Falcon 3 direct electron detector. All other cryo-EM images were recorded on Krios G1, Krios G2, and Krios G4 (Thermo Fisher Scientific) electron microscopes. Images on the Krios G1 microscope were recorded using a Gatan K3 and a Gatan energy filter with a slit width of 20 eV. Images on the Krios G2 were recorded on a Falcon-4i Camera (Thermo Fisher Scientific). Images on the Krios G4 were recorded on a Falcon-4i camera and a Selectris X (Thermo Fisher Scientific) energy filter with a slit width of 10 eV. Images were recorded at a dose of 30–40 electrons/$\text{Å}^2$, using EPU software (Thermo Fisher Scientific). Images from the Gatan K3 were saved in tiff. Images on the Falcon 4i camera were saved as EER movies and converted to tiff format using an EER grouping of 34 or 40 frames, to give a dose fractionation of approximately 1 electron/$\text{Å}^2$.

## Cryo-EM image processing

Raw micrograph movies were gain-corrected, aligned, and dose-weighted using RELION's motion correction (*Zivanov et al., 2019*). Contrast transfer function (CTF) parameters were estimated using CTFFIND-4.1 (*Rohou and Grigorieff, 2015*). Helical reconstruction was performed in RELION-5.0. Filaments were picked manually or automatically using a modified version of Topaz (*Lövestam and Scheres, 2022*). Picked particles were extracted in box sizes of 1024, 768, or 512 pixels and

down-scaled to 256, 128, or 64 pixels for initial 2D classifications. We performed reference-free 2D classification for 35 iterations with 150–200 classes, ignoring the CTF until the first peak, to assess overall filament quality, the presence of different polymorphs, and crossover distances. Polymorphs were separated by the hierarchical clustering approach (*Lövestam et al., 2024*) and quantified by counting the number of segments in the filament clusters that were identified by this method. Additional 2D classifications were run for each identified cluster, iterating the procedure until homogenous populations of 2D classes were generated. Particles that did not give rise to 2D class averages with suitable filaments were discarded in the quantifications of polymorphs. Initial 3D references were generated by 2D class averages using relion_helix_inimodel2d (*Scheres, 2020*). Selected particles were re-extracted in boxes of 384 pixels for initial 3D refinement. Subsequently, 3D classifications and 3D auto-refinements were used to optimise helical parameters and improve the reconstructions, applying symmetry where necessary. Bayesian polishing and CTF refinements were used to increase resolutions (*Zivanov et al., 2020*; *Zivanov et al., 2019*). Final maps were sharpened using standard post-processing procedures in RELION, and reported resolutions were estimated using a threshold of 0.143 in the Fourier shell correlation (FSC) between two independently refined half-maps (*Chen et al., 2013*; *Scheres and Chen, 2012*). Further details of cryo-EM structure determination are given in *Table 2*.

## Nuclear magnetic resonance

Unless otherwise stated, tau construct datasets were collected at 278 K using a Bruker Avance II+ spectrometer operating at a 700 MHz proton frequency and fitted with a 5 mm TCI triple resonance cryoprobe. All samples were prepared in 50 mM potassium phosphate (pH 7.4) with 150 mM NaCl, 10 mM DTT, and 5% $D_2O$ as a lock solvent. Backbone NH, N, Cα, Cβ, and C' resonances of tau151–391, tau297–441 with and without phosphomimetic mutations and tau297–441 Δ392–395 were assigned using isotopically enriched ($^{15}N/^{13}C$) 300 μM samples. Standard 3D datasets were acquired as pairs to provide own and preceding carbon connectivities, using between 18% and 39% non-uniform sampling (NUS) to aid faster data acquisition. Both the HNCO & HN(CA)CO, HNCA & HN(CO)CA, and CBCA(CO)NH & HNCACB complimentary 3D datasets were collected with 2048, 80, and 128 complex points in the $^1H$, $^{15}N$, and $^{13}C$ dimensions, respectively. Additional $^{15}N$ connectivities were established with an (H)N(COCA)NNH experiment with 2048, 80, and 128 complex points in the direct $^1H$ and indirect $^{15}N$ dimensions, respectively. C'-detect experiments including pulse sequences c_hcacon_ia3d and c_hcanco_ia3d (Bruker) were also acquired to facilitate backbone assignment. Data were collected with 1024, 64, and 128 points in the direct $^{13}C$, $^{15}N$, and indirect $^{13}C$ dimensions, respectively. These experiments benefitted, in the case of the 151–391 PTM samples, from the increased sensitivity of a 700 MHz TXO, X-detect optimised cryoprobe.

All raw NMR data were processed using Topspin versions 3.2 or 4 (Bruker) or, if required, NMRPipe (*Delaglio et al., 1995*) with compressed sensing for reconstruction of NUS data (*Kazimierczuk and Orekhov, 2011*) and were analysed using NMRFAM-Sparky or POKY (*Lee et al., 2021*) and MARS (*Jung and Zweckstetter, 2004*). Secondary structure preferences of individual tau residues were derived from a secondary chemical shift analysis. Random coil Cα and Cβ chemical shift values for the tau constructs' primary sequence were calculated according to *Kjaergaard et al., 2011*; *Kjaergaard and Poulsen, 2011*; *Schwarzinger et al., 2001*, with appropriate corrections for the same experimental conditions (pH and temperature). Subtraction of these values from the experimentally derived values yielded ΔCα and ΔCβ. When ΔCβ is subtracted from ΔCα, a negative value indicates that the residue resides in an extended backbone conformation, whereas positive values suggest a helical preference.

The change in relative peak position as a result of the phosphomimetic mutations (CSP) was determined using the following equation:

$$CSP = \sqrt{\left(\Delta\delta^{15}N / 5\right)^2 + \left(\Delta\delta^1H\right)^2}$$

To monitor any relative changes in peak intensities between tau constructs, peak heights extracted from processed HSQC spectra were normalised to the carboxy-terminal residue of each construct, and all other relative peak intensities were adjusted accordingly. The picosecond timescale backbone

dynamic properties of each tau construct were probed with interleaved 2D HetNOE experiments (Bruker) with a recovery delay of 5 s.

## Acknowledgements

We thank Jake Grimmett, Toby Darling, and Ivan Clayson for help with high-performance computing; David Li and Max Wilkinson for helpful discussions; and Tony Crowther for critical reading of the manuscript. This work was supported by the facilities for Biophysics, Electron Microscopy, NMR and Scientific Computing of the Medical Research Council (MRC) Laboratory of Molecular Biology. This work was supported by the MRC, as part of United Kingdom Research and Innovation (UKRI) (MC_U105184291 to MG and MC_UP_A025-1013 to SHWS). JS was funded by a Churchill Fellowship.

## Additional information

### Competing interests

Sofia Lövestam, Michel Goedert: is a named co-inventor on a patent about the technology described in this paper. Sjors HW Scheres: Reviewing editor, eLife; is a named co-inventor on a patent about the technology described in this paper. The other authors declare that no competing interests exist.

### Funding

| Funder | Grant reference number | Author |
| --- | --- | --- |
| Medical Research Council | MC_UP_A025-1013 | Sjors HW Scheres |
| Medical Research Council | MC_U105184291 | Michel Goedert |
| Churchill Fellowship | | Jenny Shi |

The funders had no role in study design, data collection and interpretation, or the decision to submit the work for publication.

### Author contributions

Sofia Lövestam, Conceptualization, Formal analysis, Investigation, Methodology, Writing – original draft, Writing – review and editing; Jane L Wagstaff, Stefan MV Freund, Formal analysis, Investigation, Methodology, Writing – review and editing; Taxiarchis Katsinelos, Jenny Shi, Formal analysis, Investigation, Writing – review and editing; Michel Goedert, Conceptualization, Supervision, Writing – original draft, Project administration, Writing – review and editing; Sjors HW Scheres, Conceptualization, Software, Formal analysis, Supervision, Methodology, Writing – original draft, Project administration, Writing – review and editing

### Author ORCIDs

Sofia Lövestam https://orcid.org/0000-0002-2152-1476
Michel Goedert https://orcid.org/0000-0002-5214-7886
Sjors HW Scheres https://orcid.org/0000-0002-0462-6540

Reviewer #1 (Public review): https://doi.org/10.7554/eLife.104778.4.sa1
Reviewer #2 (Public review): https://doi.org/10.7554/eLife.104778.4.sa2
Author response https://doi.org/10.7554/eLife.104778.4.sa3

## Additional files

### Supplementary files

MDAR checklist

## Data availability

PAD12 constructs and tau reporter cells fare available upon request. Cryo-EM maps and atomic models for PHFs formed with a mixture of 0N3R:0N4R PAD12 tau and for PHFs formed with 0N3R PAD12 tau and seeded with filaments extracted from the brain of an individual with AD have been deposited in EMDB and the PDB (see *Table 2*). NMR data (HSQC spectra plus our assignments) have been deposited to the BMRB (accession codes: 52694 – tau297–441 wt; 52695 – tau297–441 PAD-12; 52696 – tau151–391 wt; 52697 – tau151–391 PAD-12; and 53230 – tau297–441 Δ392–395). Scripts for counting of nuclei and protein aggregates are available from Zenodo, under DOIs 10.5281/zenodo.18236965 and 10.5281/zenodo.18236945, respectively.

The following datasets were generated:

| Author(s) | Year | Dataset title | Dataset URL | Database and Identifier |
|---|---|---|---|---|
| Lovestam S, Scheres SHW, Goedert M | 2024 | PAD12 0N3R:0N4R tau PHF | https://doi.org/10.2210/pdb9h5g/pdb | Worldwide Protein Data Bank, 10.2210/pdb9h5g/pdb |
| Lovestam S, Scheres SHW, Goedert M | 2024 | PAD12 0N3R tau PHF seeded by AD | https://doi.org/10.2210/pdb9h5j/pdb | Worldwide Protein Data Bank, 10.2210/pdb9h5j/pdb |
| Lovestam S, Scheres SHW, Goedert M | 2024 | 297-441 delta392-395 tau filaments | https://doi.org/10.2210/pdb9s2b/pdb | Worldwide Protein Data Bank, 10.2210/pdb9s2b/pdb |
| Lovestam S, Wagstaff J, Freund S, Scheres S | 2026 | Tau297-441 wt | https://doi.org/10.13018/BMR52694 | Biological Magnetic Resonance Data Bank, 10.13018/BMR52694 |
| Lovestam S, Wagstaff J, Freund S, Scheres S | 2026 | Tau297-441 PAD12 | https://doi.org/10.13018/BMR52695 | Biological Magnetic Resonance Data Bank, 10.13018/BMR52695 |
| Lovestam S, Wagstaff J, Freund S, Scheres S | 2026 | Tau 151-391 WT | https://doi.org/10.13018/BMR52696 | Biological Magnetic Resonance Data Bank, 10.13018/BMR52696 |
| Lovestam S, Wagstaff J, Freund S, Scheres S | 2026 | Tau151-391 PAD12 | https://doi.org/10.13018/BMR52697 | Biological Magnetic Resonance Data Bank, 10.13018/BMR52697 |
| Lovestam S, Wagstaff J, Freund S, Scheres S | 2026 | Tau297-441 delta | https://doi.org/10.13018/BMR53230 | Biological Magnetic Resonance Data Bank, 10.13018/BMR53230 |

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
