## [Editor Report · eLife Assessment]

This manuscript describes the identification and characterization of 12 specific phosphomimetic mutations in the recombinant full-length human tau protein that trigger tau to form fibrils. This **fundamental** study will allow in vitro mechanistic investigations. The presented evidence is **convincing**. This manuscript will be of interest to all scientists in the amyloid formation field.

---

## [Referee Report · Reviewer #1 (Public review)]

Summary and Strengths:

The very well-written manuscript by Lövestam et al. from the Scheres/Goedert groups entitled "Twelve phosphomimetic mutations induce the assembly of recombinant full-length human tau into paired helical filaments" demonstrates the in vitro production of the so-called paired helical filament Alzheimer's disease (AD) polymorph fold of tau amyloids through the introduction of 12 point mutations that attempt to mimic the disease-associated hyper-phosphorylation of tau. The presented work is very important because it enables disease-related scientific work, including seeded amyloid replication in cells, to be performed in vitro using recombinant-expressed tau protein.

Comments on revised version:

The manuscript is significantly improved, as also indicated by Reviewer 2, with the 100% formation of the PHF and the additional experiments to elucidate on the potential mechanism by the PTMs. This is a great work.

---

## [Referee Report · Reviewer #2 (Public review)]

Summary:

This manuscript addresses an important impediment in the field of Alzheimer's disease (AD) and tauapathy research by showing that 12 specific phosphomimetic mutations in full-length tau allow the protein to aggregate into fibrils with the AD fold and the fold of chronic traumatic encephalopathy fibrils in vitro. The paper presents comprehensive structural and cell based seeding data indicating the improvement of their approach over previous in vitro attempts on non-full-length tau constructs. The main weaknesses of this work results from the fact that only up to 70% of the tau fibrils form the desired fibril polymorphs. In addition, some of the figures are of low quality and confusing.

Strengths:

This study provides significant progress towards a very important and timely topic in the amyloid community, namely the in vitro production of tau fibrils found in patients.

The 12 specific phosphomimetic mutations presented in this work will have an immediate impact in the field since they can be easily reproduced.

Multiple high-resolution structures support the success of the phosphomimetic mutation approach.

Additional data show the seeding efficiency of the resulting fibrils, their reduced tendency to bundle, and their ability to be labeled without affecting core structure or seeding capability.

Comments on revised version:

Generally, I am satisfied with the revisions. Specifically, the new results showing 100% formation of PHF is a significant improvement.

---

## [Author Response]

The following is the authors’ response to the previous reviews.

**Reviewer #1:**
The manuscript is significantly improved, as also indicated by Reviewer 2, with the 100% formation of the PHF and the additional experiments to elucidate on the potential mechanism by the PTMs. This is a great work.
**Reviewer #2:**
One (minor) issue I do still have is how confusingly the NMR data are presented. Although the authors revised Figure 6 and added labels to the HSQCs etc., this figure and its supplements are still very hard to understand. I think this can be easily fixed by highlighting in the figures and also figure captions which changes/differences the reader is supposed to appreciate and why.

We have added labelling to Figure 6 and extended the legends to its Supplements.

After our fist revision, the level of evidence in the eLife assessment was described as *convincing*. In our opinion the results in this paper, which include 11 cryo-EM data sets and NMR experiments on 6 tau constructs among other data, provide a level of evidence that extends beyond the state-of-the-art in the field.